# A four-DNA methylation biomarker is a superior predictor of survival of patients with cutaneous melanoma

Wenna Guo[1†], Liucun Zhu[2†], Rui Zhu[2], Qihan Chen[1], Qiang Wang[1]*, Jian-Qun Chen[1]*

[1]State Key Laboratory of Pharmaceutical Biotechnology, School of Life Sciences, Nanjing University, Nanjing, China; [2]School of Life Sciences, Shanghai University, Shanghai, China

**Abstract** Cutaneous melanoma (CM) is a life-threatening form of skin cancer. Prognostic biomarkers can reliably stratify patients at initial melanoma diagnosis according to risk, and may inform clinical decisions. Here, we performed a retrospective, cohort-based study analyzing genome-wide DNA methylation of 461 patients with CM from the TCGA database. Cox regression analyses were conducted to establish a four-DNA methylation signature that was significantly associated with the overall survival (OS) of patients with CM, and that was validated in an independent cohort. Corresponding Kaplan–Meier analysis displayed a distinct separation in OS. The ROC analysis confirmed that the predictive signature performed well. Notably, this signature exhibited much higher predictive accuracy in comparison with known biomarkers. This signature was significantly correlated with immune checkpoint blockade (ICB) immunotherapy-related signatures, and may have potential as a guide for measures of responsiveness to ICB immunotherapy.
DOI: https://doi.org/10.7554/eLife.44310.001

*For correspondence:
wangq@nju.edu.cn (QW);
chenjq@nju.edu.cn (J-QC)

†These authors contributed equally to this work

Competing interests: The authors declare that no competing interests exist.

## Introduction

Cutaneous melanoma (CM) is a malignant neoplasia characterized by rapid progression, metastasis to regional lymph nodes as well as distant organs, and limited responsiveness to therapeutics (*MacKie et al., 2009*). It contributes to more than 80% death of skin cancer patients (*Miller and Mihm, 2006*), and its incidence is one of the most rapidly increasing cancers in the United States where it is estimated that there will be 91,270 new cases and 9,320 deaths due to this disease in 2018 (*Siegel et al., 2018*). Although significant progress has been made in both understanding CM biology and genetics, and in therapeutic approaches, the prognosis remains poor due to the high potential for CM invasion and metastasis. This malignancy still represents a major public health problem worldwide. Primary localized melanoma has a relatively high survival rate. The 5- year survival rate prognosis of patients with lymph node or distant metastases was only 15–20% (*Siegel et al., 2018*; *Weiss et al., 2015*). Assessment of patients prior to therapy may aid in the identification of individuals at high risk for recurrence and may guide the development of future targeted treatment strategies. For example, for patients who are at high operative risk, conservative therapy may be helpful in the operative decision-making process. Molecular biomarkers could provide additional prognostic information and insight into the mechanisms of melanoma progression, as well as guide treatment selection. Consequently, new strategies for the identification of effective prognostic biomarkers may improve the clinical management of melanoma by providing more accurate prognostic information.

Aberrant DNA methylation is an epigenetic hallmark of cancer and actively contributes to cancer development and progression by inactivating tumor suppressor genes (*Egger et al., 2004*). Some characteristics of methylation markers render them particularly attractive for development of clinically applicable biomarkers, including high stability in biologic samples, limited susceptibility to tumor environmental factors, and ease of detection (*Keeley et al., 2013*). A growing number of studies have demonstrated that aberrant DNA methylation plays important roles in tumorigenesis and progression, and DNA methylation has great potential to act as a biomarker for predicting the prognosis of patients with a variety of malignancies (*Egger et al., 2004*; *Guo et al., 2004*; *Roh et al., 2016*). For instance, *GATA4* is epigenetically silenced in gastrointestinal cancers (*Akiyama et al., 2003*) and lung cancer (*Guo et al., 2004*); *OPCML* promoter methylation has been identified as a biomarker for predicting ovarian cancer prognosis (*Zhou et al., 2014*); *PCDH19* methylation is associated with poor hepatocellular carcinoma prognosis (*Zhang et al., 2018*) and Sigalotti *et al*. identified a 17-gene methylation signature as a molecular marker of survival in stage IIIC melanoma (*Sigalotti et al., 2012*). Unfortunately, many recent melanoma studies have several limitations including relatively small sample cohorts, lack of subsequent biomarker validation, narrow focus on patient specimens with specific clinical features, or investigation of only one or a few genes. These studies lack the comprehensive and systematic approach of genome-wide methylation analysis. Because of this, the identified methylation signatures do not have universal prognostic power, and their capability is limited by the specimen type. The Cancer Genome Atlas (TCGA) database (*Hudson et al., 2010*), a large well-annotated database with nearly 500 CM samples collected worldwide is a helpful resource for developing promising biomarkers with prospective studies into a methylation signature which can be of clinical utility.

Consequently, the purpose of this study was to identify DNA methylation biomarkers so as to explore the utility of DNA methylation analysis for cancer prognosis. The whole genome methylation profiles of tumor tissues from patients with CM in the TCGA database were analyzed, and the potential clinical significance of methylation biomarkers serving as molecular prognostic predictors was examined using the Kaplan–Meier method and receiver operating characteristic (ROC) analyses. Furthermore, the prognostic capacity of the identified methylation biomarker was evaluated with an independent cohort of patients in the Gene Expression Omnibus (GEO) database. Also, we investigated the independence and reproducibility in various clinical groups, as well as the possible role of the methylation biomarker in immune-checkpoint blockade (ICB) immunotherapy.

## Results

### Clinical characteristics of the study populations

The study was conducted on 461 CM patients who are clinically and pathologically diagnosed with CM. Of these patients, 286 (62.04%) were male and 175 (37.96%) were female. The median age at diagnosis and Breslow thickness of these patients were 58 years (range, 15–90) and 3.0 mm (range, 0–75 mm), respectively, and the median OS were 1,827 days. In regard to tumor tissue site, the regional lymph node was the most common site, followed by primary tumor, regional cutaneous or subcutaneous metastatic tissue and distant metastasis. The pathologic stage was defined according to the American Joint Committee on Cancer (AJCC) Cancer staging manual, and 6 (1.30%), 75 (16.27%), 139 (30.15%), 171 (37.097%) and 23 (4.99%) patients were in stage 0, I, II, III and IV, respectively. Anatomic sites were located at various positions of the patients, including head and neck, extremity and trunk, and the extremities were the most common location (42.08%). Ulceration occurs in 167 patients, and only 26.68% (*N* = 123) of patients received postoperative or adjuvant chemotherapy. The distribution and selected demographic characteristics of melanoma patients are summarized in *Table 1*.

### Derivation of prognostic DNA methylation markers from the training cohort

By subjecting the DNA methylation level data in the training cohort to univariate Cox proportional hazard regression analysis, a total of 4454 DNA methylation sites that significantly (p<0.001) correlated with the OS of patients with CM were identified as candidate markers. Subsequently, these candidate markers were used to perform multivariate Cox stepwise regression analyses, and a

**Table 1.** Clinicopathological characteristics of CM patients from TCGA database.

| Characteristics | Groups | Patients | | | | | | |
|---|---|---|---|---|---|---|---|---|
| | | Total (N = 461) | | Training cohort (N = 307) | | Validation cohort (N = 154) | | |
| | | No | % | No | % | No | % | |
| Sex | Male | 286 | 62.04 | 195 | 63.52 | 89 | 57.79 | |
| | Female | 175 | 37.96 | 112 | 36.48 | 65 | 42.21 | |
| Age at diagnosis | Median | 58 | | 58 | | 58 | | |
| | Range | 15–90 | | 15–90 | | 19–90 | | |
| | ≤58 | 234 | 50.76 | 154 | 50.16 | 80 | 51.95 | |
| | >58 | 227 | 49.24 | 153 | 49.84 | 74 | 48.05 | |
| Tumor tissue site | Primary tumor | 104 | 22.56 | 76 | 24.76 | 28 | 18.18 | |
| | Regional cutaneous or subcutaneous tissue | 73 | 15.84 | 51 | 16.61 | 22 | 14.29 | |
| | Regional lymph node metastasis | 216 | 46.85 | 154 | 50.16 | 62 | 40.26 | |
| | Distant metastasis | 65 | 14.10 | 23 | 7.49 | 42 | 27.27 | |
| | Unknown | 3 | 0.65 | 3 | 0.98 | 0 | 0.00 | |
| Pathologic stage | 0 | 6 | 1.30 | 5 | 1.63 | 1 | 0.65 | |
| | I | 75 | 16.27 | 53 | 17.26 | 22 | 14.29 | |
| | II | 139 | 30.15 | 92 | 29.97 | 47 | 30.52 | |
| | III | 171 | 37.09 | 117 | 38.11 | 54 | 35.06 | |
| | IV | 23 | 4.99 | 13 | 4.23 | 10 | 6.49 | |
| | Unknown | 47 | 10.20 | 27 | 8.79 | 20 | 12.99 | |
| Anatomic site | Head and neck | 36 | 7.81 | 21 | 6.84 | 15 | 9.74 | |
| | Extremity | 194 | 42.08 | 129 | 42.02 | 65 | 42.21 | |
| | Trunk | 167 | 36.23 | 117 | 38.11 | 50 | 32.47 | |
| | Others/Unknown | 64 | 13.88 | 40 | 13.03 | 24 | 15.58 | |
| Breslow thickness (mm) | <2 | 126 | 27.33 | 85 | 27.69 | 41 | 26.62 | |
| | 2–5 | 124 | 26.90 | 79 | 25.73 | 45 | 29.22 | |
| | >5 | 106 | 22.99 | 78 | 25.41 | 28 | 18.18 | |
| | Unknown | 105 | 22.78 | 65 | 21.17 | 40 | 25.97 | |
| Ulceration | Present | 167 | 36.23 | 120 | 39.09 | 47 | 30.52 | |
| | Absent | 145 | 31.45 | 100 | 32.57 | 45 | 29.22 | |
| | NA/Unknown | 149 | 32.32 | 87 | 28.34 | 62 | 40.26 | |
| Chemotherapy | Yes | 123 | 26.68 | 70 | 22.80 | 53 | 34.42 | |
| | NO | 319 | 69.20 | 227 | 73.94 | 92 | 59.74 | |
| | Unknown | 19 | 4.12 | 10 | 3.26 | 9 | 5.84 | |
| Vital Status | Alive | 241 | 52.28 | 167 | 54.40 | 74 | 48.05 | |
| | Dead | 220 | 47.72 | 140 | 45.60 | 80 | 51.95 | |

DOI: https://doi.org/10.7554/eLife.44310.002

hazard ratio model consisting of four methylation sites (cg06778853, cg24670442, cg18456782, cg26263675) was selected as the optimum prognostic model for predicting OS. The risk score formula based on the DNA methylation level and regression coefficients of four methylation sites was created as follows: Risk score = $-1.912 \times \beta$ value of cg06778853 $+4.262 \times \beta$ value of cg24670442 $+1.229 \times \beta$ value of cg18456782 $- 2.108 \times \beta$ value of cg26263675. Among these methylation sites, cg24670442 and cg18456782 had positive coefficients, indicating a correlation between higher DNA methylation level and shorter OS, while higher levels of DNA methylation in

cg06778853 and cg26263675 sites correlated with longer OS. The genes corresponding with these four sites were *KLHL21* (kelch like family member 21), *GBP5* (guanylate binding protein 5), *OCA2* (OCA2 melanosomal transmembrane protein), and *RAB37* (RAB37, member RAS oncogene family). The list of these four DNA methylation sites, their chromosomal locations, their *P* values and coefficients obtained in Cox regression analysis, are shown in *Supplementary file 1*.

Meanwhile, for these four DNA methylation sites, the DNA methylation level between patients exhibiting long-term (>5 years) and short-term (<5 years) survival was significantly different (*Figure 1A*) ($p < 0.001$, Mann–Whitney *U* test). Patients exhibiting long-term survival tended to have lower methylation levels of cg24670442, cg18456782 and higher methylation levels of the other two methylation sites, consistent with the results of multivariate Cox regression analysis. Moreover, principal component analysis (PCA) was carried out using four methylation values at selected biomarkers (*Figure 1—figure supplement 1*). The difference of PC1 and PC4 is 15.42%, indicating the continuous capturing of information. And the combination of four methylation markers can effectively distinguish patients with long- and short-term survival.

## Association between the four-DNA methylation signature and patient OS in training and validation cohorts

According to the results of Cox regression analysis, the four-DNA methylation signature were significantly associated with the OS of patients using the risk scores as a continuous variable in both the training and validation cohorts (training cohort: p=1.73E-7, HR: 2.72, 95% CI of HR: 1.91–3.88; validation cohort: p=3.19E-6, HR: 3.58, 95% CI of HR: 2.04–6.29). To determine the potential predictive value of this four-DNA methylation signature in the prognosis, Kaplan–Meier curves along with the Wilcoxon test were used to visualize and compare the OS of patients in the low- versus high-risk group which were classified using the median risk score (3.69) of the training cohort as the cutoff point, and the distribution of the risk predictor scores for the training and validation cohort was illustrated in *Figure 1—figure supplement 2*. As expected, the survival of patients in the low-risk group was significantly improved in comparison with patients in the high-risk group (*Figure 1B* for training cohort, median OS of 2,639 and 1,119 days, respectively, and *Figure 1C* for validation cohort, median OS of 2,306 and 1,297 days, respectively). These results confirmed that the novel four-DNA methylation signature could successfully stratify patients into high- and low-risk groups, implying its significance in determining CM prognosis.

## The prognostic potential of the four-DNA methylation signature

To understand the specificity of the four-DNA methylation signature in predicting survival, the AUC values of the ROC curves were calculated by time-dependent ROC analysis using a categorical variable for OS < 5 years compared with the signature. For this purpose, patients for whom we did not have at least 5 years of follow-up were excluded, unless death had been documented. In both training and validation cohorts, the four-DNA methylation signature has good discriminatory capacity for predicting patient OS, with dynamic AUC estimates exceeding 0.80 (*Figure 2A*) and 0.75 (*Figure 2B*) in training and validation cohorts, respectively. These results indicate that the four-DNA methylation signature has high sensitivity and specificity, and has great potential to serve as a prognostic biomarker in clinical applications.

## Evaluation of the four-DNA methylation signature for OS prediction in an independent cohort

To further examine the prognostic values of the four-DNA methylation signature in another independent cohort, Kaplan–Meier and ROC analyses were carried out in an independent cohort (GSE51547, N = 47). Similarly, patients with high or low risk were grouped based on the median risk score of the training cohort. The results showed that the four-DNA methylation performed well, and patients in the low-risk group had a significantly longer OS than those in the high-risk group (p<0.05) (*Figure 2C*). Here, due to the limited sample size of the independent cohort with follow up times more than 5 years (N = 2, 4.25%), the AUC was calculated using 1 year as the cutoff (N = 32, 68.08%), and the AUC estimate was 0.708 (p=0.022, 95% CI: 0.54–0.88) (*Figure 2D*), suggesting that the four-DNA methylation signature can also predict the survival of CM patients in other independent cohorts.

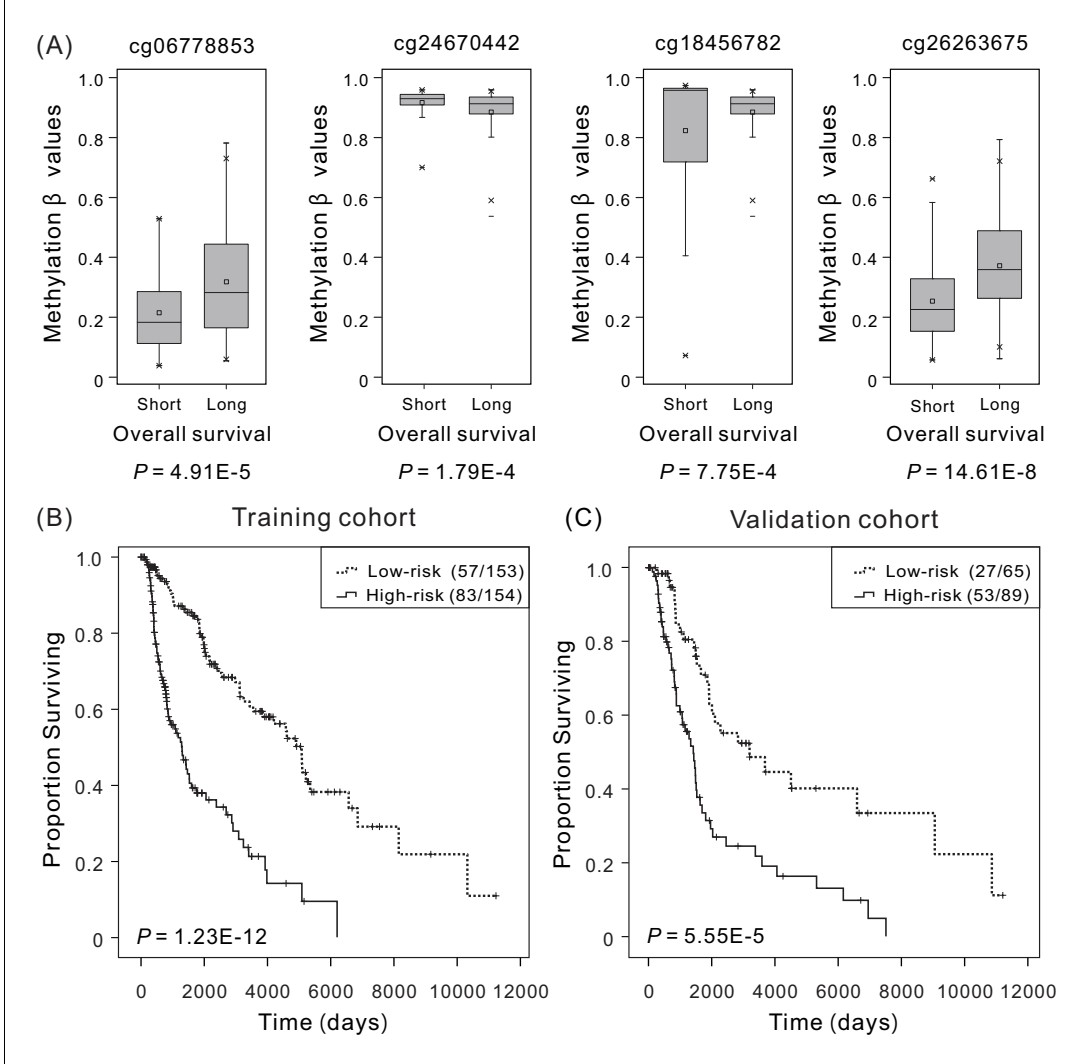

**Figure 1.** Overall survival (OS) and methylation levels of patient cohorts. (A) Methylation β values of samples from patients with short survival (OS <5 years) and long survival (OS >5 years) in the training cohort. Within each methylation site, the thick line represents the median value, the bottom and top of the boxes are the 25th and 75th percentiles (interquartile range). The whiskers encompass 1.5 times the interquartile range. The difference between short and long survival groups was compared through the Mann–Whitney $U$ test, and $P$ values are shown below the plots. The Kaplan–Meier curves along with the Wilcoxon test were used to visualize and compare the OS of the low-risk versus high-risk groups in the training cohort ($N = 307$) (B) and the validation cohort ($N = 154$) (C). Here 'low-risk (57/153)' refers to that a total of 153 patients in the low-risk group, in which 57 with last clinical status 'death', and 'low-risk (83/154)' refers to that a total of 154 patients in the high-risk group, in which 83 with last clinical status 'death'. It can be concluded that higher risk scores are significantly associated with worse OS (p<0.001).

DOI: https://doi.org/10.7554/eLife.44310.003

The following source data and figure supplements are available for figure 1:

**Source data 1.** The distribution of overall survival (OS) and methylation levels for patient in the training cohort.

DOI: https://doi.org/10.7554/eLife.44310.007

**Source data 2.** The OS of patients in the low-risk versus high-risk groups for the training cohort (N = 307).

DOI: https://doi.org/10.7554/eLife.44310.008

**Source data 3.** The OS of patients in the low-risk versus high-risk groups for the validation cohort (N = 154).

DOI: https://doi.org/10.7554/eLife.44310.009

**Figure supplement 1.** The principal component analysis (PCA) models were constructed using four methylation values at selected biomarkers.

DOI: https://doi.org/10.7554/eLife.44310.004

**Figure supplement 2.** Distribution of the four-methylation risk predictor score values in the training cohort and the validation cohort.

DOI: https://doi.org/10.7554/eLife.44310.005

*Figure 1 continued on next page*

Figure 1 continued

**Figure supplement 3.** Correlation between the expression of the genes and their methylation levels was evaluated for each gene through the Pearson's correlation test.

DOI: https://doi.org/10.7554/eLife.44310.006

## Independence of the four-DNA methylation signature in the OS prediction from clinical and pathological factors

An important feature of a good prognostic signature is that it should be independent or additive to currently used clinicopathologic prognostic factors. Clinical and pathological characteristics, such as patients' age, sex, AJCC stage, tumor thickness and ulceration status also have been considered to be the predominant predictors used to determine prognosis of melanoma patients. To assess the independence and applicability of this four-DNA methylation signature, patients were regrouped according to different clinicopathological characteristics. Over the last few decades, the incidence of CM has been increasing rapidly in males compared to females of all ages, with the exception of young women who appear to be at higher risk than young men (*Robsahm et al., 2013*). The incidence in male patients is 1.6 times higher than that of female patients, and regrouping was performed based on patients' sexes and ages at initial diagnosis in the following way: age $\leq 50$ ($N$ = 141, 30.58%), 50 < age $\leq$ 70 ($N$ = 202, 43.82%), and age >70 ($N$ = 118, 25.60%). Irrespective of sexes and ages, Kaplan–Meier curves showed that patients in the low-risk group had significantly (p<0.001) longer OS, and the AUC values were more than 0.75 (*Figure 3* and *Figure 3—figure supplement 1*), suggesting that the four-DNA methylation signature is independent of patient sex and age. Considering that once the tumor metastasizes to distant tissues, the 5 year survival rate is very low (*Siegel et al., 2018*), we regrouped patients based on the site of sample obtainment, including distant metastasis, subcutaneous tissue, and regional lymph node metastasis. Kaplan–Meier and ROC analyses demonstrated that the survival of patients in low-risk groups was much improved in comparison with patients in high-risk groups, and the four-DNA methylation signature had high predictive performance (*Figure 3—figure supplement 2*). Meanwhile, research has shown that DNA methylation changes in relation to disease stage (*Wouters et al., 2017*), and survival outcomes can vary widely even at a single stage (*Weiss et al., 2015*). Because of limited sample size at each stage, patients were separated into early-stage (0 and I and II) and advanced-stage (III and IV) cohorts. Despite the markedly different outcomes in terms of the extent of disease, the OS between high- and low-risk groups are significantly (p<0.001) different, and the AUC in early-stage and advanced-stage cohorts were 0.814 and 0.809, respectively (*Figure 3—figure supplement 3*). Furthermore, whether the tumor was located in head and neck or extremity or trunk, the four-DNA methylation signature performed well in differentiating low- and high-risk groups, and patients in high-risk groups showed a trend towards worse OS (*Figure 3—figure supplement 4*).

Considering that Breslow thickness is the strongest prognostic factor in CM, patients who have Breslow thickness more than 2 mm are at the greatest risk of developing locoregional cutaneous metastases (*Messeguer et al., 2013*), we investigated whether the four-DNA methylation signature could classify patients with different survival risk for patients with different Breslow thickness. The results indicated that the four-DNA methylation signature was effective in distinguishing the high-risk patients from low-risk patients for patients of any Breslow thickness groups (*Figure 3—figure supplement 5*). CM ulceration status has also been shown in many studies to be a major and independent prognostic parameter. Regardless of ulceration, four-DNA methylation signature proved useful for identifying patients with low risk (*Figure 3—figure supplement 6*). Additionally, we found no association between the predictive performance of the four-DNA methylation signature and whether a patient received adjuvant chemotherapy (*Figure 3—figure supplement 7*). All these results indicated that the four-DNA methylation signature provides a better reference for different regrouped cohorts owing to the effectiveness of risk stratification, suggesting that the signature was an independent applicable prognostic predictor of patient survival. The results of Kaplan–Meier and ROC analyses are summarized in *Table 2*.

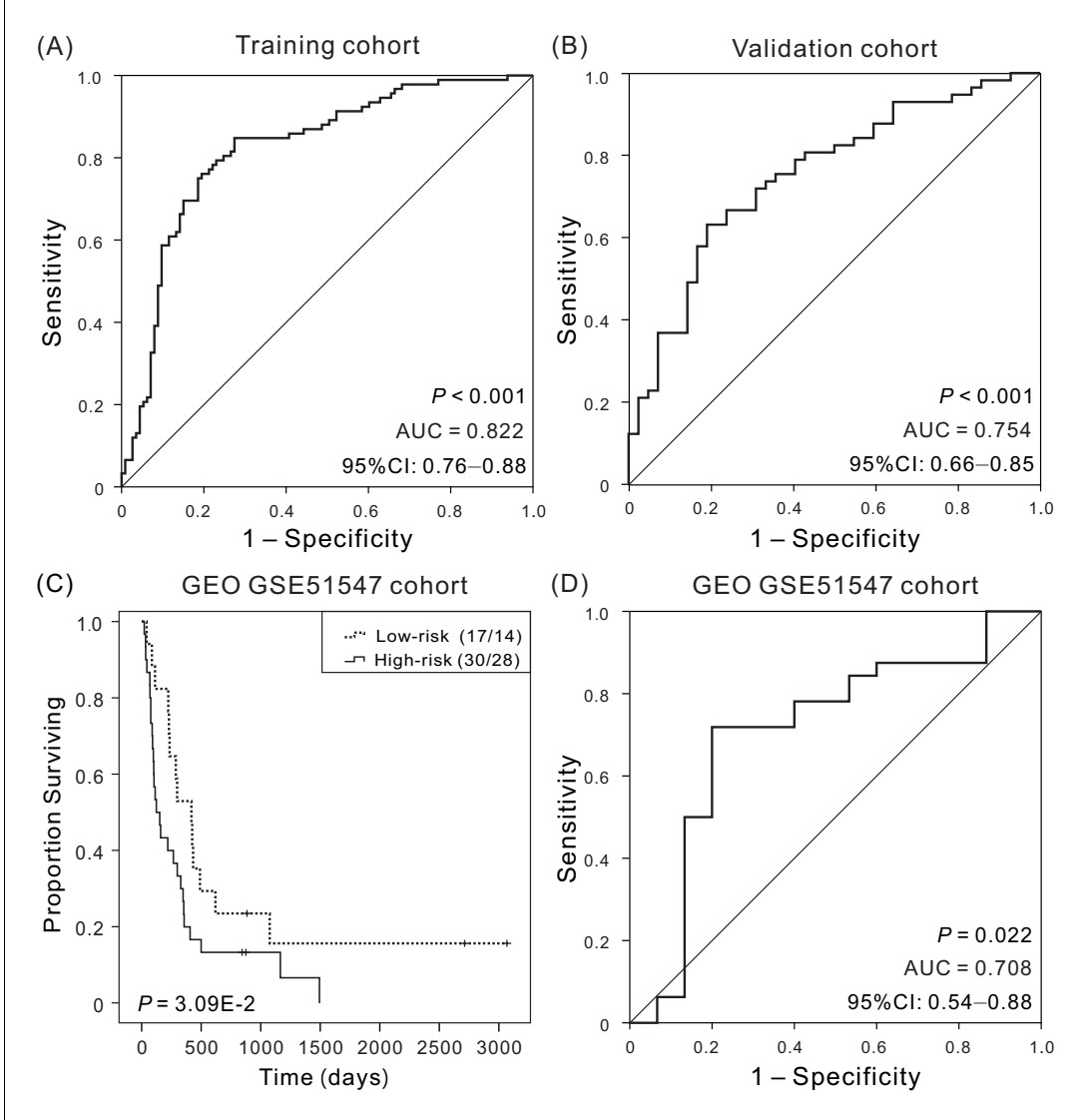

**Figure 2.** Kaplan–Meier and ROC analyzes of the four-DNA methylation signature in predicting the OS of patients. (**A**) ROC analysis of sensitivity and specificity of the four-DNA methylation signature in predicting the OS of patients in training cohort, with an AUC of 0.822 (**B**) ROC analysis in validation cohort, with an AUC of 0.754. (**C**) Kaplan–Meier survival curves demonstrating the correlation between the four-DNA methylation signature and poorer OS of patients in an independent cohort (GSE51547). (**D**) ROC curves show the sensitivity and specificity of the signature in predicting the patient' OS, AUC = 0.708.

DOI: https://doi.org/10.7554/eLife.44310.010

The following source data and figure supplements are available for figure 2:

**Source data 1.** The OS and four-DNA methylation risk scores of patients in training cohort.
DOI: https://doi.org/10.7554/eLife.44310.014

**Source data 2.** The OS and four-DNA methylation risk scores of patients in validation cohort.
DOI: https://doi.org/10.7554/eLife.44310.015

**Source data 3.** The OS of patients in the low-risk versus high-risk groups for an independent cohort (GSE51547).
DOI: https://doi.org/10.7554/eLife.44310.016

**Source data 4.** The OS and four-DNA methylation risk scores of patients in an independent cohort (GSE51547).
DOI: https://doi.org/10.7554/eLife.44310.017

**Figure supplement 1.** Kaplan–Meier and ROC analyses of individual DNA methylation in the TCGA validation cohort.
DOI: https://doi.org/10.7554/eLife.44310.011

**Figure supplement 2.** Kaplan–Meier and ROC analyses of patients with CM in both training cohort and validation cohort.
DOI: https://doi.org/10.7554/eLife.44310.012

*Figure 2 continued on next page*

*Figure 2 continued*

**Figure supplement 3.** Kaplan–Meier and ROC analyses of individual DNA methylation in the training cohort.
DOI: https://doi.org/10.7554/eLife.44310.013

## Comparison of the four-DNA methylation signature with other known prognostic biomarkers

In addition, numerous prognostic markers have previously been identified for CM, utilizing archival tumor tissues or single institutional studies. *MITF* has been identified as a lineage survival oncogene amplified in malignant melanoma (*Garraway et al., 2005*), *CTLA-4* expression has been shown to be associated with a more favorable prognosis of patients with CM (*Goltz et al., 2018*), and *CD74* was identified as a useful prognostic marker associated with OS of patients in stage III melanoma (*Ekmekcioglu et al., 2016*). The presence of *MGMT* promoter methylation is an independent variable associated with longer OS (*Cesinaro et al., 2012*), and the methylation of *PTEN* has been reported as an independent negative prognostic factor in terms of OS (*Roh et al., 2016*). To determine whether our signature has superior ability to predict patient survival, compared with known biomarkers, ROC analyses of other biomarkers were carried out in the validation cohort (*Figure 4A*) and independent cohort (*Figure 4B*), in the same way our four-DNA methylation signature was analyzed. The results demonstrated that the four-DNA methylation signature had higher AUC than all the other known biomarkers in both validation cohort and independent cohort. The results of ROC analysis are shown in *Figure 4A* and *Supplementary file 2*, revealing that the four-DNA methylation signature was a superior predictor, and provided better stability and reliability in predicting the OS of patients with CM.

## Association of the four-DNA methylation signature with ICB immunotherapy-related signature

In recent years, cancer immunotherapy using ICB has created a paradigm shift in the treatment of advanced-stage cancers, and provides significant clinical benefits for patients with CM (*Hugo et al., 2016*). ICB treatment primarily targets programmed cell death 1 (PD-1), programmed cell death-ligand 1 (PD-L1), and cytotoxic T-lymphocyte-associated protein 4 (CTLA-4) (*Sharma et al., 2017*). Programmed cell death-ligand 2 (PD-L2) was found to play a role in the regulation of anti-tumor immunity (*Umezu et al., 2019*). Multiple studies have shown that tumor mutational burden (TMB) may be a surrogate for overall neoantigen load, and it is correlated with clinical benefit from multiple checkpoint inhibitors (*Rizvi et al., 2015*). Features of the tumor microenvironment (TME) also were associated with the response to ICB therapy (*Jeschke et al., 2017*; *Riaz et al., 2017*). Although TME is determined by both DNA methylation and gene expression, DNA methylation may reflect distributions of cell subtypes more adequately, given that the relationship of only 2 DNA molecules per cell is of a more linear nature than are thousands of mRNA copies exposed to mRNA degradation (*Jeschke et al., 2017*). Indeed, Jeschke *et al* have identified a five-DNA methylation signature of tumor-infiltrating lymphocytes (MeTIL), which could more accurately measure TIL distributions in a sensitive manner and predict survival and tumor immune responses than gene expression-based immune markers (*Jeschke et al., 2017*). Additionally, the tumor immune response is increasingly recognized to be associated with better clinical outcomes (*Cristescu et al., 2018*; *Goltz et al., 2018*; *Jeschke et al., 2017*). Here we investigated the prognostic impact of these immunotherapy-related signatures in the validation cohort (*Figure 4C*). To investigate the possible role of our four-DNA methylation signature in ICB treatment, we performed one-to-one correlation between these known immunotherapy-related signatures and our signature. As expected, *PD-1*, *PD-L1*, *PD-L2*, and *CTLA-4* mRNA were coexpressed (p<0.001) (*Figure 4D*), which is consistent with the reported coexpression of immune checkpoint molecules (*Goltz et al., 2018*). TMB was not significantly correlated with any other signature, which is also consistent with previous reports (*Cristescu et al., 2018*). Surprisingly, our four-DNA methylation signature was significantly negatively correlated with *PD-1*, *PD-L1*, *PD-L2*, and *CTLA-4* (p<0.05 and $r$ = –0.485,–0.338, –0.322, and –0.131, respectively); meanwhile, MeTIL was significantly negatively correlated with *PD-1*, *PD-L1*, and *PD-L2* (p<0.05 and $r$ = –0.411,–0.288, –0.288, respectively), but was not significantly correlated with *CTLA-4* (p=0.233 and $r$ = –0.056) (*Figure 4C*), suggesting that some elements or all of the four-DNA methylation signature may play a

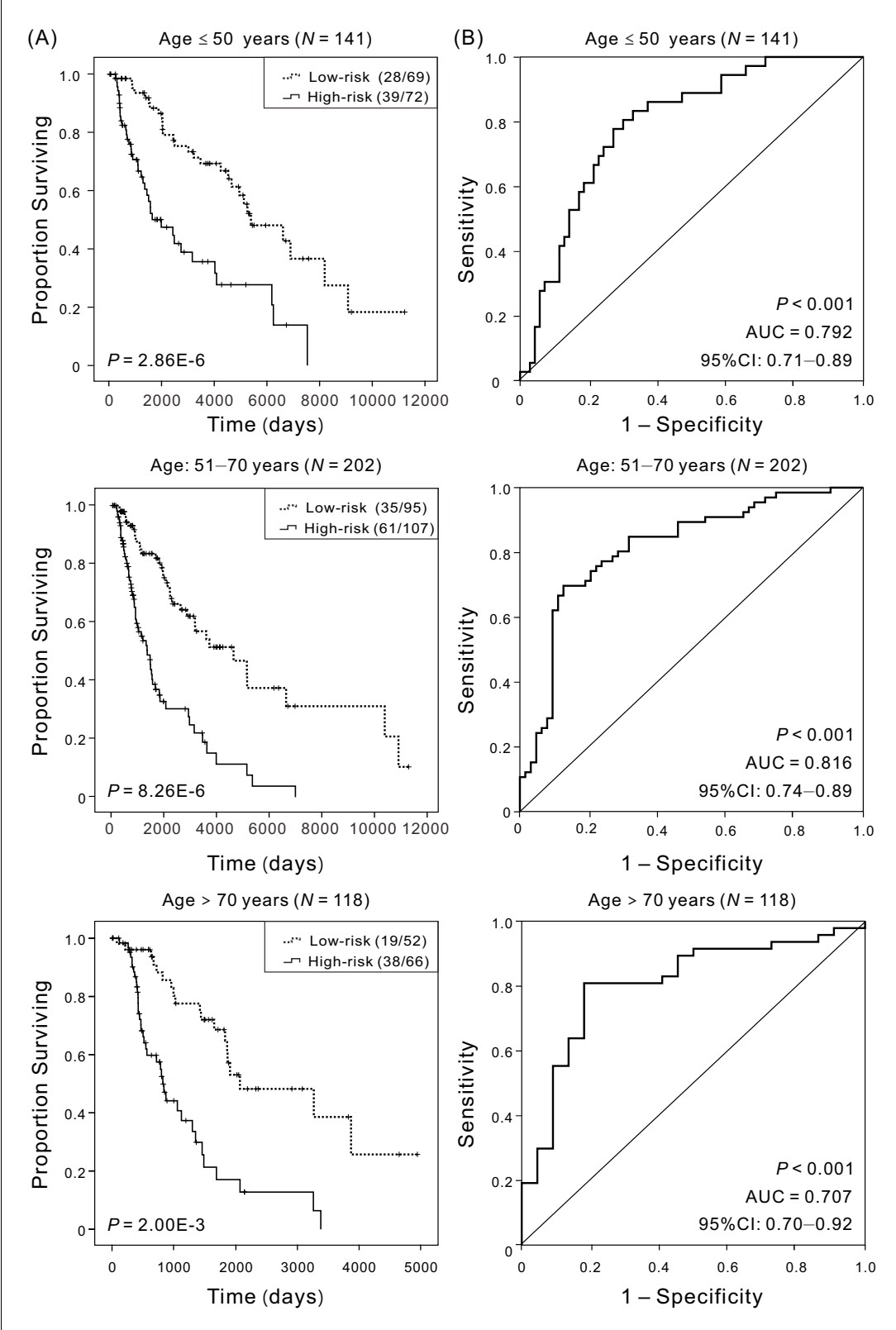

**Figure 3.** Kaplan–Meier and ROC analyses of patients with CM in different age cohorts, grouping based on their ages at initial diagnosis:≤50 (N = 141, 30.58%), 51–70 (N = 202, 43.82%),>70 (N = 118, 25.60%), respectively. (A) Kaplan–Meier analysis with Wilcoxon test was performed to estimate the differences in OS between the low-risk and high-risk patients. (B) ROC curves of the four-DNA methylation signature were used to demonstrate the sensitivity and specificity in predicting the OS of patients.

*Figure 3 continued on next page*

*Figure 3 continued*

DOI: https://doi.org/10.7554/eLife.44310.018

The following source data and figure supplements are available for figure 3:

**Source data 1.** The OS of patients in the low-risk versus high-risk groups for patients with different age at initial diagnosis.
DOI: https://doi.org/10.7554/eLife.44310.026
**Source data 2.** The OS and four-DNA methylation risk scores of patients with different age at initial diagnosis.
DOI: https://doi.org/10.7554/eLife.44310.027
**Figure supplement 1.** Kaplan–Meier and ROC analyses of CM patients in different sex groups.
DOI: https://doi.org/10.7554/eLife.44310.019
**Figure supplement 2.** Kaplan–Meier and ROC analyses of CM patients with tumor from different tissue sites.
DOI: https://doi.org/10.7554/eLife.44310.020
**Figure supplement 3.** Kaplan–Meier and ROC analyses of CM patients in early stage cohorts (stage 0 and I and II, *N* = 231) and advanced stage (stage III and IV, *N* = 194).
DOI: https://doi.org/10.7554/eLife.44310.021
**Figure supplement 4.** Kaplan–Meier and ROC analyses of CM patients with tumor from different anatomic sites.
DOI: https://doi.org/10.7554/eLife.44310.022
**Figure supplement 5.** Kaplan–Meier and ROC analyses of CM patients with Breslow thickness, including <2 mm, 2–5 mm, and >5 mm.
DOI: https://doi.org/10.7554/eLife.44310.023
**Figure supplement 6.** Kaplan–Meier and ROC analyses of CM patients with ulceration or no ulceration, respectively.
DOI: https://doi.org/10.7554/eLife.44310.024
**Figure supplement 7.** Kaplan–Meier and ROC analyses of CM patients received adjuvant chemotherapy or not, respectively.
DOI: https://doi.org/10.7554/eLife.44310.025

role in measures of responsiveness to ICB immunotherapy. In fact, cg24670442, one of the four-DNA methylation signatures, corresponding to *GBP5* gene, was significantly negatively correlated with *PD-1*, *PD-L1*, *PD-L2*, *CTLA-4* ($p < 0.001$, and $r = -0.681, -0.405, -0.436, -0.171$, respectively), and

**Table 2.** Results of Kaplan–Meier and ROC analyses based on various regrouping methods.

| Regrouping factors | Group | Sample size | Kaplan–Meier *P*-value | AUC | 95% CI of AUC |
|---|---|---|---|---|---|
| Sex | Male | 286 | 4.69E-10 | 0.786 | 0.72–0.85 |
| | Female | 175 | 2.82E-08 | 0.844 | 0.77–0.92 |
| Age at diagnosis | ≤50 | 141 | 3.46E-06 | 0.792 | 0.71–0.88 |
| | 51–70 | 202 | 1.14E-07 | 0.816 | 0.74–0.89 |
| | >70 | 118 | 1.75E-05 | 0.807 | 0.70–0.92 |
| Tumor metastasis site | Distant metastasis | 65 | 2.63E-03 | 0.836 | 0.73–0.94 |
| | Regional cutaneous or subcutaneous tissue | 73 | 1.79E-09 | 0.859 | 0.76–0.96 |
| | Regional lymph node metastasis | 216 | 2.01E-07 | 0.749 | 0.67–0.83 |
| Pathologic stage | 0 and I and II | 231 | 1.56E-11 | 0.814 | 0.75–0.88 |
| | III and IV | 194 | 6.75E-07 | 0.809 | 0.73–0.89 |
| Anatomic site | Head and neck | 36 | 7.76E-05 | 0.960 | 0.00–1.00 |
| | Extremity | 194 | 9.28E-06 | 0.787 | 0.71–0.87 |
| | Trunk | 167 | 3.87E-06 | 0.756 | 0.66–0.85 |
| Breslow thickness (mm) | <2 | 126 | 1.83E-06 | 0.830 | 0.74–0.92 |
| | 2–5 | 124 | 1.79E-04 | 0.742 | 0.64–0.85 |
| | >5 | 106 | 1.31E-02 | 0.808 | 0.65–0.96 |
| Ulceration | Present | 167 | 6.84E-05 | 0.809 | 0.69–0.93 |
| | Absent | 145 | 1.79E-04 | 0.736 | 0.64–0.83 |
| Chemotherapy | Yes | 123 | 5.46E-06 | 0.791 | 0.69–0.79 |
| | NO | 319 | 1.10E-10 | 0.797 | 0.74–0.86 |

DOI: https://doi.org/10.7554/eLife.44310.028

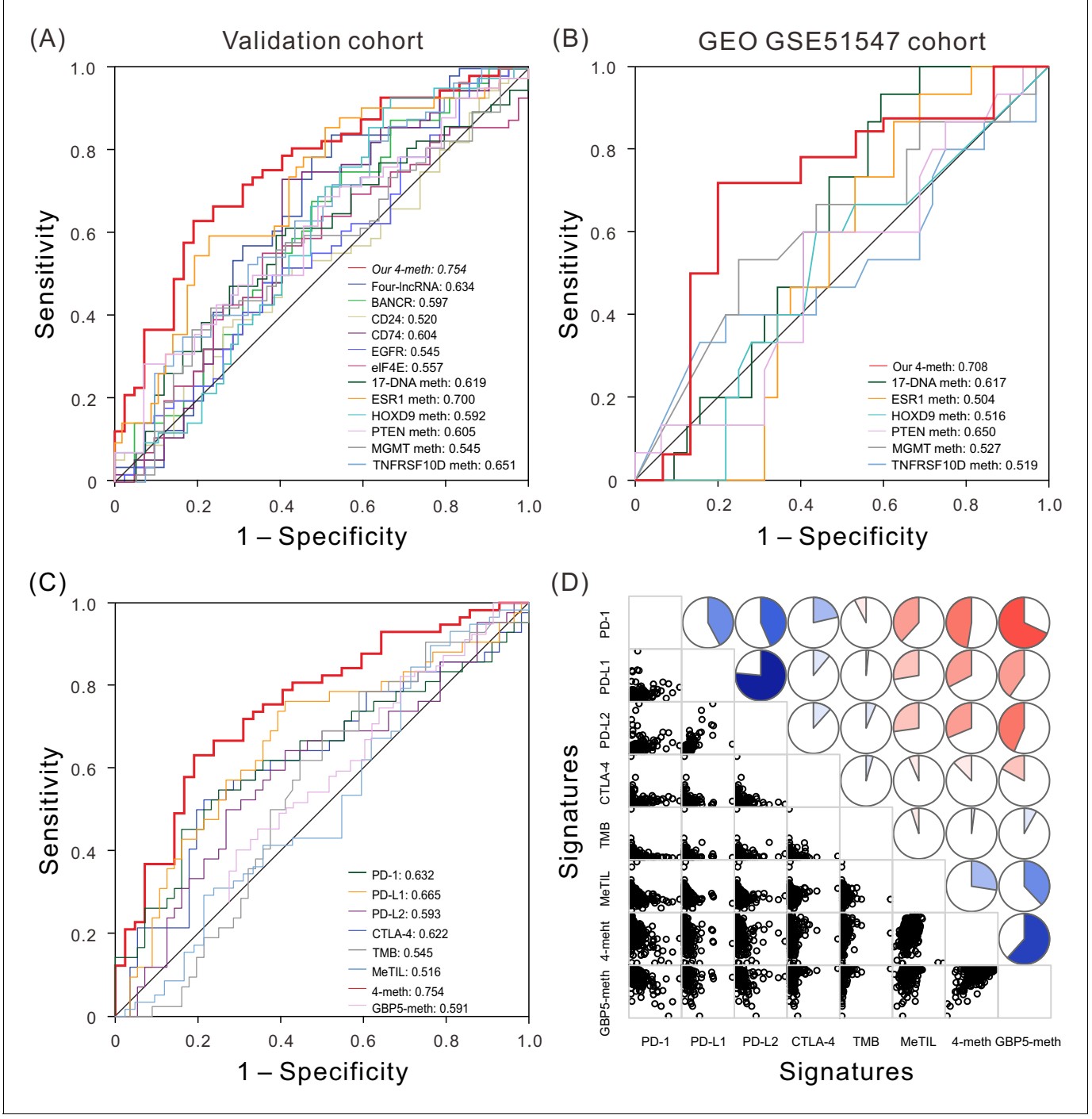

**Figure 4.** ROC and correlation analyses of different prognostic biomarkers. (**A**) ROC curves show the sensitivity and specificity of our four-DNA methylation signature and other known biomarkers in predicting the OS of patients from TCGA validation dataset. (**B**) ROC curves of our four-DNA methylation signature and other known biomarkers in predicting the OS of patients from another independent cohort. (**C**) ROC curves of our four-DNA methylation signature, *GBP5*-meth, and known immune checkpoint genes, TMB and MeTIL in predicting the OS of patients from TCGA validation dataset. (**D**) Correlation analyses between known immune checkpoint genes, TMB and MeTIL, our four-DNA methylation signature, as well as *GBP5*-meth, one of our four DNA methylation sites. Lower triangle: scatter plots showing the correlation between two signatures. Upper triangle: circle symbols represent the one-to-one correlation coefficient; each correlation coefficient is shown by fill area and intensity of shading, which increases uniformly as the correlation value moves away from 0; blue for positive correlation, red for negative correlation.

DOI: https://doi.org/10.7554/eLife.44310.029

The following source data is available for figure 4:

*Figure 4 continued on next page*

*Figure 4 continued*

**Source data 1.** The OS and expression level or methylation level of the four-methylation signature and other known biomarkers in validation cohort.
DOI: https://doi.org/10.7554/eLife.44310.030
**Source data 2.** The OS and methylation level of the four-methylation signature and other known biomarkers in an independent cohort (GSE51547).
DOI: https://doi.org/10.7554/eLife.44310.031
**Source data 3.** The OS, our four-DNA methylation signature, GBP5-meth, and known immune checkpoint genes, TMB and MeTIL for patients for patients from TCGA validation dataset.
DOI: https://doi.org/10.7554/eLife.44310.032
**Source data 4.** The four-DNA methylation signature, GBP5-meth, and known immune checkpoint genes, TMB and MeTIL for patients from TCGA dataset.
DOI: https://doi.org/10.7554/eLife.44310.033

patients with long-term survival exhibiting lower methylation levels and higher expression level. Meanwhile, *GBP5* has been proved to be induced by interferon-gamma (IFN-γ), and could serve as the substitute indicator of IFN-γ, which promotes not only immunomodulation but also anticancer activity, and play a role in immunotherapy (*Chang et al., 2015*; *Yamamoto et al., 2012*). The raw *P* values of Pearson correlation and Bonferroni correction adjusted *P* values between our four-DNA methylation signature and other known signatures are shown in *Supplementary file 3*. Collectively, these results imply that our four-DNA methylation signature, although developed to accurately stratify patients in terms of prognosis, may also play a role in ICB immunotherapy.

## Discussion

CM is the deadliest form of skin cancer and lead to 60,712 deaths in 2018 (*Miller and Mihm, 2006*). In recent years, the importance of DNA methylation in the biology of CM has been increasingly acknowledged. For instance, hypermethylated estrogen receptor alpha (ER-α) is a significant factor in melanoma progression (*Mori et al., 2006*); methylation-dependent *SOX9* expression mediates invasion in human melanoma cells and is a negative prognostic factor in advanced melanoma (*Cheng et al., 2015*), and *MGMT* gene promoter methylation in metastatic CM is associated with longer survival (*Cesinaro et al., 2012*). However, these studies usually concentrated on single gene methylation or patient samples with specific clinical characteristics, and combinations of DNA methylation as biomarkers could achieve higher sensitivity and specificity than individual DNA methylation (*Dai et al., 2011*). Moreover, a limited overlap between signatures published by various studies can be observed, possibly due to variation in disease causes, tissue heterogeneity, or stage of disease at the point of analysis (*Tremante et al., 2012*). The TCGA database provides a large number of samples with a variety of clinical characteristics. Based on a TCGA dataset that included 461 CM samples, the current study identified a prognostic DNA methylation signature with potential clinical applicability, validated it in an independent cohort, and investigated its high reproducibility and utility in various clinical groups.

CM has a high degree of heterogeneity in terms of clinical, dermatological, and histopathological presentation (*Coricovac et al., 2018*). Several parameters, such as age, sex, stage of disease, Breslow thickness, and ulceration status have significant influence on CM patient prognosis, in which Breslow thickness has been demonstrated as the most important clinicopathological characteristic for predicting prognosis. In order to be clinically useful, a DNA methylation signature must be independent of clinical factors. Here we adopted a grouping method not influenced by our subjective, that is based on the TCGA series number of patients, without adjusting for clinical parameters. If detection, treatment and surgical excision occur when the tumor burden is restricted to a primary site, a patient's prognostic survival is enhanced, but once the disease has metastasized to distant organs such as the brain and liver, prognosis is poor (*Balch et al., 2009*). Since patients with primary tumors had shorter follow-ups, none more than 5 years, we analyzed the independence and applicability of our four-DNA methylation signature in samples obtained from distant metastasis, subcutaneous tissue, and regional lymph node metastasis. The results showed that our signature was independent of tumor metastatic sites. Moreover, as patients suffering from early stages of the disease exhibit higher potential for healing, a prognostic signature that can also efficiently risk-stratify these patients would have higher clinical application value (*Segura et al., 2010*). Our signature has been demonstrated to be not only independent of all clinical factors, but also to have higher

predictive performance for patients with tumor thickness of less than 2 mm and patients in early stages (AUCs were 0.830 and 0.814, respectively). Therefore, our signature was an independent applicable prognostic biomarker, which may be of high clinical value.

Considering that an ideal prognostic marker is one that can also efficiently risk-stratify in other independent cohorts, we employed GEO dataset (GSE51547) to further evaluate the practicality of our four-DNA methylation signature. Although the predictive accuracy in the GEO dataset is not as high as in the validation dataset due to the low number of samples ($N = 47$), the four-DNA methylation signature performed well in distinguishing low- and high-risk groups (AUC = 0.708, p<0.05). Furthermore, it was demonstrated that in both the validation and independent cohorts, our signature outperformed other known prognostic biomarkers, including mRNA, lncRNA, and DNA methylation, and statistical comparison using Z-test revealed that it has significantly higher (p<0.05) predictive performance than almost all the other known biomarkers. When further samples become available it will be important to analyze this methylation signature in another validation dataset.

Targeting immune checkpoints such as PD-1, PD-L1, and CTLA-4 have achieved noteworthy benefit in multiple cancers by blocking immunoinhibitory signals and enabling patients to produce an effective antitumor response, especially in patients with CM (*Riaz et al., 2017*). However, a significant limitation of ICB is that less than one-third of patients respond to ICB treatment, and identification of ICB response biomarkers and resistance regulators is a critical challenge (*Sharma et al., 2017*). DNA methylation plays a critical role in cell lineage specification and may serve as a specific molecular marker for measurement of immune responses. Recently, Jeschke *et al* highlighted the power of MeTIL to evaluate local and functional TIL-based tumor immune responses and the ability of this approach to improve prognosis (*Jeschke et al., 2017*). However, the identification of MeTIL was focused on CpGs that are highly differentially methylated between T lymphocytes and epithelial cells. Lymphocytes only account for a small fraction of TME (*Pretscher et al., 2009*); thus, there may be bias when using MeTIL as a prognostic marker to predict survival outcomes. Intriguingly, the correlation analyses and the observed predictive performance suggested that our four-DNA methylation signature was significantly correlated with the ICB immunotherapy-related signature. Additionally, our signature displayed higher predictive performance than other known signatures, including *PD-1*, *PD-L1*, *PD-L2*, *CTLA-4*, and MeTIL. These results demonstrate that our four-DNA methylation signature, although developed for accurate prognosis, may also have potential as a guide for precision cancer ICB immunotherapy.

Furthermore, epigenetic changes have been shown to alter gene expression, and epigenetic inactivation of tumor suppressor genes has been implicated in tumorigenesis of various malignancies, including CM (*Herman and Baylin, 2003*). Here, the expression of *GBP5* and *KLHL21* were significantly (p<0.001) negatively correlated with their methylation levels, and the other two genes show significant positive correlation (p<0.001) between the expression and their methylation levels (*Figure 1—figure supplement 3*). We also found that expression of this four-gene can also be used as a prognostic biomarker (*Figure 2—figure supplement 1*), but the four-DNA methylation biomarker offer a better potential to fulfill much more sensitive and specific prognostic test. For our four-DNA methylation sites, researchers have revealed that their corresponding genes may be crucial in immunity and cancer development, including CM. For instance, *GBP5* promotes NLRP3 inflammasome assembly and immunity in mammals (*Shenoy et al., 2012*). *GBP5* was induced by IFN-γ, could serve as a marker of IFN-γ-induced classically activated macrophages and the substitute indicator of IFN-γ, which can directly suppress tumorigenesis and infection and/or can modulate the immunological status in cancer cells (*Chang et al., 2015*; *Yamamoto et al., 2012*). Meanwhile, *GBP5* expression in CM is associated with favorable prognosis (*Wang et al., 2018*). *RAB37*, as a tumor suppressor gene, promotes M1-like macrophage infiltration and suppresses tumor growth (*Tzeng et al., 2018*), and it was frequently down-regulated due to promoter hypermethylation in metastatic lung cancer, can serve as a potential predictive biomarker (*Wu et al., 2009*). *RAB37*-mediated *SFRP1* secretion suppresses cancer stemness, and dysregulated RAB37-SFRP1 pathway confers cancer stemness via the activation of Wnt signaling (*Cho et al., 2018*). *OCA2* is involved in the melanin biosynthetic process and mammalian pigmentation (*Crawford et al., 2017*), and the DNA variant in intron of *OCA2* (rs4778138) has been found associated with CM risk (*Law et al., 2015*). The hypomethylation levels of cg18456782 (*OCA2*) was associated with lower expression of *OCA2* and a lower risk. Meanwhile separating CM patients by median expression of *OCA2*, there is a significant differential survival (p<0.0001) with low expression favoring better survival. All these results suggest a risk pattern for

*OCA2* gene in CM. *KLHL21* could affect cell migration and invasion, play an essential role in tumorigenesis and progression, and it might serve as a potential therapeutic target for cholangiocarcinoma (*Chen et al., 2018*) and hepatocellular carcinoma (*Shi et al., 2016*). Although the functional mechanism of these four genes in CM still needs further study, significant correlation between these four genes and OS or response to therapy of patients with CM, and DNA methylation might also be suitable as biomarkers for response to ICB therapy.

In addition, the results of the univariate Cox regression, Kaplan–Meier and ROC analyses for the four individual methylation sites show that each of the DNA methylation sites also could distinguish high- and low-risk patients, but the predictive performances were lower than the combination of these four DNA methylation sites in both the training and validation cohorts (*Figure 2—figure supplements 2–3*), indicating that single methylation sites are likely to play a role in the prognostic prediction, and a combination of methylation sites might offer better potential to fulfill much more sensitive and specific prognostic tests for patients with CM. To the best of our knowledge, the prognostic value of this multi-marker signature in melanoma has not been previously reported. Therefore, the current study provides new insight that a combination of epigenetic biomarkers could help to improve risk-stratification and prediction of survival in patients with melanoma.

In conclusion, we identified and verified a four-DNA methylation signature that was significantly associated with the OS of patients in TCGA and an independent cohort. The four-DNA methylation signature was not only independent of clinical factors including patient sex, age, stage, tumor location, and Breslow thickness, but also exhibited superior ability in predicting OS compared with known biomarkers. The four-DNA methylation signature was able to stratify patients with startling accuracy in survival differences, suggesting that it may be used to select patients for therapies, and help to determine whether patients may require more or less aggressive treatment. Furthermore, the four-DNA methylation signature was significantly correlated with the ICB immunotherapy-related signature. Therefore, though these exploratory findings are warranted to validate the potential role of this prognostic signature in clinical application and the functional characterization in CM development, these four-DNA methylation sites, or some of them, may participate in the progress of the cancer, and have great potential implications for both risk-stratification, adjuvant management and measures of response to ICB immunotherapy of patients with CM.

## Materials and methods

### DNA methylation data of CM tissues

The DNA methylation data and related clinical information of patients with CM were downloaded from the TCGA database (*Hudson et al., 2010*). TCGA DNA methylation data (level 3) were obtained using Infinium Human Methylation 450 BeadChip (Illumina Inc, CA, USA). For each CpG site, the ratio of fluorescent signal was measured by that of a methylated probe relative to the sum of the methylated and unmethylated probes, a ratio termed β value, also known as DNA methylation level. β values were standardized and assigned a value from 0 (no methylation) to 1 (100% methylation). Only the data corresponding to patients for whom clinical survival information was available were selected. The correlation between DNA methylation levels and corresponding survival in CM was analyzed. Overall, 461 samples with 485,577 DNA methylation sites were analyzed in this study. According to the TCGA series number, these samples were divided into two cohorts: the first two-thirds were used as the training cohort for identifying and constructing prognostic biomarkers, and the remaining one-third were used as a validation cohort for verifying the predictive performance of the biomarker. Detailed patient eligibility information have been described in the previous study (*Cancer Genome Atlas Network, 2015*), and the following clinicopathological parameters relevant to this study were selected from the TCGA clinical patient data files to perform analyses: sex, age at diagnosis, tumor tissue site, Breslow thickness, pathologic stage, ulceration status, and last clinical status. The number of samples used from each cohort are shown in *Table 1*. Also, an additional methylation dataset and corresponding clinical data were downloaded from the GEO database (47 patients, GEO accession number: GSE51547) and used as an independent validation cohort.

## Statistical analyses

As we previously reported (*Chen et al., 2017*), the outcome of interest was death due to CM, and overall survival (OS) was defined as the time from the date of a patient's diagnosis to the date of CM-related death or last follow-up. The univariate Cox proportional hazard analysis was first conducted in the training cohort to identify methylation markers significantly ($p < 0.001$) associated with patient survival. Then the variables significantly associated with OS in univariate analysis were included in multivariate Cox regression analysis and constructed models comprising all possible combinations of two to five factors, aiming at further selecting combined biomarker correlated with OS. Hazard ratios (HR) and corresponding 95% confidence interval (CI) were assessed using Cox proportional hazard models. The linear combination of model predictors weighted by regression coefficients was defined as the risk predictive formula and was used to predict the survival risk of patients. Briefly, according to the formula, the risk score for each patient was calculated, and the patients were then classified into high- or low-risk groups using the median risk score as the cutoff value. Secondly, the Kaplan–Meier survival curves were drawn using the 'survival' package to visualize the cumulative survival time displaying number of patients at risk for some time points, and Wilcoxon rank test was used to assess the differences in OS between the two groups with the consideration that it is a more sensitive measure than the log-rank test to compare differences in survival probability between groups that occur in early points in time (*Youden, 1950*). Lastly, the risk scores were also evaluated for utility in predicting OS of patients by identifying the area under the Receiver operating characteristic (ROC) curve (AUC) along with 95% CI in the ROC analysis, which was conducted with the 'pROC' package. Additionally, Pearson correlation coefficients were calculated to characterize the potential association between DNA methylation level and gene expression level. All statistical calculations were carried out using the R statistical environment (R version 3.4.4).

## Additional information

### Funding

| Funder | Grant reference number | Author |
|---|---|---|
| National Natural Science Foundation of China | 31471200 | Qiang Wang |
| National Natural Science Foundation of China | 31501045 | Liucun Zhu |
| National Natural Science Foundation of China | 31741073 | Qihan Chen |
| Nanjing University | Graduated Research and Innovation Fund (No. 2017CL06) | Wenna Guo |

The funders had no role in study design, data collection and interpretation, or the decision to submit the work for publication.

### Author contributions

Wenna Guo, Data curation, Software, Formal analysis, Investigation, Methodology, Writing—original draft, Writing—review and editing; Liucun Zhu, Software, Formal analysis, Validation, Investigation, Methodology, Writing—review and editing; Rui Zhu, Validation, Investigation, Writing—review and editing; Qihan Chen, Funding acquisition, Validation, Writing—review and editing; Qiang Wang, Conceptualization, Software, Supervision, Funding acquisition, Validation, Project administration, Writing—review and editing; Jian-Qun Chen, Conceptualization, Supervision, Validation, Project administration, Writing—review and editing

### Author ORCIDs

Qiang Wang [ORCID] http://orcid.org/0000-0003-2907-9851

### Decision letter and Author response

Decision letter https://doi.org/10.7554/eLife.44310.045

Author response https://doi.org/10.7554/eLife.44310.046

## Additional files

### Supplementary files

• Source code 1. Univariate Cox proportional hazard analysis source code.
DOI: https://doi.org/10.7554/eLife.44310.034

• Source code 2. Multivariate Cox proportional hazard analysis source code.
DOI: https://doi.org/10.7554/eLife.44310.035

• Supplementary file 1. Four significantly survival-related methylation sites in training dataset.
DOI: https://doi.org/10.7554/eLife.44310.036

• Supplementary file 2. The ROC results of four-DNA methylation signature and other known biomarkers.
DOI: https://doi.org/10.7554/eLife.44310.037

• Supplementary file 3. The correlation raw P values and adjusted P values with the Bonferroni correction between our 4-meth signature and other known signatures.
DOI: https://doi.org/10.7554/eLife.44310.038

• Transparent reporting form
DOI: https://doi.org/10.7554/eLife.44310.039

### Data availability

We downloaded the data from publicly available databases: The Cancer Genome Atlas database (https://cancergenome.nih.gov/) and GEO database (under accession code GSE51547).

The following previously published datasets were used:

| Author(s) | Year | Dataset title | Dataset URL | Database and Identifier |
|-----------|------|---------------|-------------|-------------------------|
| International Cancer Genome C, Hudson TJ, Anderson W | 2010 | International network of cancer genome projects | https://portal.gdc.cancer.gov/projects/TCGA-SKCM | The Cancer Genome Atlas, TCGA-SKCM |
| Lauss M | 2015 | DNA methylation in melanoma is connected to gene expression phenotypes and MITF pathway expression | https://www.ncbi.nlm.nih.gov/geo/query/acc.cgi?acc=GSE51547 | NCBI Gene Expression Omnibus, GSE51547 |

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
