## [Decision Letter]

Thank you for submitting your article "A four-DNA methylation biomarker predicts survival of patients with cutaneous melanoma" for consideration by *eLife*. Your article has been reviewed by 3 peer reviewers and the evaluation has been overseen by a Reviewing Editor and Maarten van Lohuizen as the Senior Editor. The reviewers have opted to remain anonymous.

The reviewers have discussed the reviews with one another and the Reviewing Editor has drafted this decision to help you prepare a revised submission.

Summary:

In the current manuscript, Guo et al. use published data on cutaneous melanoma including DNA methylation status to develop a 4-DNA methylation biomarker that is predictive of overall survival based on historical case data.

Significant points:

1) The identification of a novel epigenetic four-DNA methylation signature that can successfully stratify patients into high- and low-risk groups, with AUC estimates exceeding 0.80 and 0.75 in training and validation cohorts.

2) Confirmation of the prognostic value of this signature in an independent (though smaller) cohort.

3) The finding that the four-DNA methylation signature was effective in distinguishing the high-risk patients from low-risk patients, independent of clinical parameters like Breslow thickness.

4) The finding that the 4-gene methylation signature is inversely correlated with expression of immune-checkpoint genes.

Essential revisions:

1) Absence of functional data: Does the methylation pattern of the 4 genes correlate with the expression of these genes? The authors are encouraged to analyze RNA Seq data from TCGA to address this issue. This issue is important because if methylation correlates with expression, then these four genes may (also) be important therapeutic targets. More specific questions to address at a basic level – Are the genes at/near the methylation sites expressed in melanomas at the RNA or protein level? Does expression (in at least a subset or other dataset) correlate with the methylation status at the loci? Some plausible roles for the genes GBP5, RAB37, etc. are offered in the discussion, but some functional analysis of these genes or the pathways hypothesized to be affected (as described in the Discussion) would substantially increase the degree to which this study could move the field forward.

2) Questions about the analysis: The authors should carefully control for clinical features from the very beginning of their analysis rather than estimating that clinical features seem to not have an effect on their predictor after the model is fully built. It is hard to evaluate and interpret any intermediate findings (are they driven by clinical biases or actual signal in methylation data?). Specifically: was univariate analysis in the training cohort adjusted for age, tumor stage/grade and tumor tissue site? To perform a search for markers independent from clinical features it seems to be critical to adjust for clinical differences first. Will this change marker selection for prediction model? The authors should introduce a comparison with known melanoma methylation signals in (e.g. MITF region, etc.) as a positive control for their model. Interestingly, they do cite and use the data from M. Lauss et al. who found the MITF signal, but never check whether their predictor-building approach captures the same signal.

Also: subsection “Statistical analysis”. "The univariate Cox proportional hazard analysis was first conducted in the training cohort to identify methylation markers significantly (P < 0.001) associated with patient survival."

How was the significance level was determined? According to the manuscript 461 samples with 485,577 DNA methylation sites were analyzed. Most-widely used Bonferroni correction (might be too aggressive cutoff, I am not insisting on using this particular one) should result in P < 0.05 / 485,577 ~ 1x10^-7^. Subsection “Derivation of prognostic DNA methylation markers from the training cohort”: 4,454 markers were included into the multivariate regression analysis (above comment on significance level determination applies).

3) Validation: The four-DNA methylation signature "beats" the other biomarkers in Supplementary file 2/Figure 4A-C. Data may not be available for all of these studies, but for those based on DNA methylation, how does the four-DNA signature compare to the datasets used to generate other methylation predictive markers (Supplementary file 2 17-DNA methylation, CTLA-4, etc.)? Presumably these had validation datasets as well (it seems unfair to expect the four-DNA signature of this study to "beat" the training set for another marker, but other validation sets should be a reasonable comparator). The authors did include a comparison using their validation set, but additional validation datasets would strengthen the generalizability of their biomarker/signature.

4) It would be helpful to include a supplemental figure/panel illustrating distribution of the 4-site methylation risk predictor score value for the TCGA melanoma cohort and then plot the threshold dividing "low-risk" and "high-risk" groups (median). Subsequently, the Figure 3 (main text) and supplementary figure legends need to be explained better (e.g. in Figure 3A "low-risk (28/69)" what does (28/69) mean? I seem to be unable to find this in the text or figure caption.)

5) One additional panel in Figure 1 (or supplement) might be helpful: principal component analysis plot using methylation values at 4 selected biomarkers that illustrates separation between "high" and "low" survival groups. This should clearly illustrate direct effect of methylation level on survival differences.

6) Subsection “Association of the four-DNA methylation signature with ICB immunotherapy-related signature”: "significantly negatively correlated with PD-1, PD-L1, PD-L2, CTLA-4 (P < 0.05 [..,]". Multiple hypothesis testing significance adjustment should be considered. Based on Figure 4D – 4-meth signature correlation was estimated against 7 other signatures (or 6 if GBP5-meth is considered a part of 4-meth), thus significance should be determined using P-value threshold P < 0.05/7 (if Bonferroni correction is used). Please, report raw p-values for this analysis in the supplement.

[Editors' note: further revisions were requested prior to acceptance, as described below.]

Thank you for resubmitting your work entitled "A four-DNA methylation biomarker is a superior predictor of survival of patients with cutaneous melanoma" for further consideration at *eLife*. Your revised article has been favorably evaluated by Maarten van Lohuizen (Senior Editor), a Reviewing Editor, and three reviewers.

The manuscript has been improved but there are some minor remaining issues that need to be addressed before acceptance, as outlined below:

The reviewers appreciate the effort made by the authors to address their concerns. The result is a strengthened manuscript.

1) Without another validation dataset (major point #3), the generalizability of the results will remain a potential weakness but publishing this study with this caveat noted in the text/discussion is of value.

It is recommended that the authors mention this, for example by adding the following sentence to the end of the third paragraph of the Discussion section: When further samples become available it will be important to analyze this methylation signature in another validation dataset.

2) The DNA variant in intron of OCA2 has been found to have a protective effect in melanoma GWAS (rs4778138, β = -0.18, PMID: 26237428). The authors state that cg18456782 (OCA2) marker "had positive coefficients, indicating a correlation between higher DNA methylation level and shorter overall survival". If one looks at the TCGA survival data and separates melanoma patients by median OCA2 expression, there is a significant differential survival (p~1e-5) with low expression favoring better survival. This is consistent with the presented data and suggests that while in GWAS the OCA2 variant has been found having a protective function, survival and expression data suggests a risk pattern for OCA2 gene.

Discussing this in the Discussion section would strengthen the manuscript because there is no sufficient experimental data on OCA2.

3) Prior to the analysis the training cohort was not adjusted for clinical parameters like age, tumor stage/grade and tumor tissue site; the authors have explained how they controlled for this, but this fact should be mentioned in the Discussion section so that results could be interpreted with the appropriate caution.

4) The Abstract should be modified as follows:

Cutaneous melanoma (CM) is a life-threatening form of skin cancer. Prognosticbiomarkers can reliably stratify patients at initial melanoma diagnosis according to risk, and may inform clinical decisions. Here, we performed a retrospective, cohort-based study analyzing genome-wide DNA methylation of 461 patients with CM from the TCGA database. Cox regression analyses were conducted to establish a four-DNA methylation signature that was significantly associated with the overall survival (OS) of patients with CM, and that was validated in an independent cohort. Corresponding Kaplan-Meier analysis displayed a distinct separation in OS. The ROC analysis confirmed that the predictive signature performed well. Notably, this signature exhibited much higher predictive accuracy in comparison with known biomarkers. This signature was significantly correlated with immune checkpoint blockade (ICB) immunotherapy-related signatures, and may have potential as a guide for measures of responsiveness to ICB immunotherapy.

---

## [Author Response]

Essential revisions:1) Absence of functional data: Does the methylation pattern of the 4 genes correlate with the expression of these genes? The authors are encouraged to analyze RNA Seq data from TCGA to address this issue. This issue is important because if methylation correlates with expression, then these four genes may (also) be important therapeutic targets. More specific questions to address at a basic level – Are the genes at/near the methylation sites expressed in melanomas at the RNA or protein level? Does expression (in at least a subset or other dataset) correlate with the methylation status at the loci? Some plausible roles for the genes GBP5, RAB37, etc. are offered in the discussion, but some functional analysis of these genes or the pathways hypothesized to be affected (as described in the Discussion) would substantially increase the degree to which this study could move the field forward.

Thank you for noting this. Indeed, it is helpful to analyze the correlation between the methylation pattern of the 4 genes and the expression of these genes. It is generally believed that differential methylation can affect gene expression by influencing transcription factor binding (Medvedeva et al., 2014). In the manuscript, we have analyzed the expression of these genes and the correlation between the expression and their methylation levels, found that all these 4 genes were expressed in melanomas, the expression of *KLHL21* and *GBP5* were significantly (*P* < 0.001) negatively correlated with their methylation levels, and the other two genes show significant positive correlation (*P* < 0.001) between the expression and their methylation levels. In fact, we have investigated the role of the expression levels of these four genes in prognostic prediction, and found that their expression levels can also be used as prognostic markers (AUC: 0.662, 95%CI: 0.55-0.77), but the four-DNA methylation biomarker offer a better potential to fulfill much more sensitive and specific prognostic test (AUC: 0.754, 95%CI: 0.66-0.85). A possible explanation is that DNA methylation not only correlated with expression of corresponding genes, but also reflected upstream regulations. Therefore, this study focused on DNA methylation prognostic biomarkers. To show these results in more detail, we made changes in our manuscript and added two supplementary figures.

In the Discussion section: “Furthermore, epigenetic changes have been shown to alter gene expression, and epigenetic inactivation of tumor suppressor genes has been implicated in tumorigenesis of various malignancies, including CM (41). Here, the expression of *GBP5* and *KLHL21* were significantly (*P* < 0.001) negatively correlated with their methylation levels, and the other two genes show significant positive correlation (*P* < 0.001) between the expression and their methylation levels (Figure 1—figure supplement 3). We also found that expression of this four-gene can also be used as a prognostic biomarker (Figure 2—figure supplement 1), but the four-DNA methylation biomarker offer a better potential to fulfill much more sensitive and specific prognostic test.”

2) Questions about the analysis: The authors should carefully control for clinical features from the very beginning of their analysis rather than estimating that clinical features seem to not have an effect on their predictor after the model is fully built. It is hard to evaluate and interpret any intermediate findings (are they driven by clinical biases or actual signal in methylation data?). Specifically: was univariate analysis in the training cohort adjusted for age, tumor stage/grade and tumor tissue site? To perform a search for markers independent from clinical features it seems to be critical to adjust for clinical differences first. Will this change marker selection for prediction model? The authors should introduce a comparison with known melanoma methylation signals in (e.g. MITF region, etc.) as a positive control for their model. Interestingly, they do cite and use the data from M. Lauss et al. who found the MITF signal, but never check whether their predictor-building approach captures the same signal.Also: subsection “Statistical analysis”. "The univariate Cox proportional hazard analysis was first conducted in the training cohort to identify methylation markers significantly (P < 0.001) associated with patient survival."How was the significance level was determined? According to the manuscript 461 samples with 485,577 DNA methylation sites were analyzed. Most-widely used Bonferroni correction (might be too aggressive cutoff, I am not insisting on using this particular one) should result in P < 0.05 / 485,577 ~ 1x10^-7^. Subsection “Derivation of prognostic DNA methylation markers from the training cohort”: 4,454 markers were included into the multivariate regression analysis (above comment on significance level determination applies).

Control for clinical features from the very beginning of the analysis may reduce or avoid the influence of clinical biases on the identification of predictor, but the clinicopathological parameters relevant to this study including sex, age at diagnosis, tumor tissue site, Breslow thickness, pathologic stage, ulceration status, and last clinical status, it was difficult to control all the clinical features from the very beginning, so we chose an grouping method based on the TCGA series number of patients, which is not influenced by our subjective. Chi-square test showed no significant difference in the distribution of patients’ sex (*P* = 0.194), age (*P* = 0.718), tumor stage (*P* = 0.597), and tumor tissue site (*P* = 0.337) between the training set and the validation set. Here we take a brute force approach, that is, exhaustively chose 2-5 sites from all DNA methylation sites that significantly (*P* < 0.001) correlated with the OS of patients as covariates in multivariate Cox regression analysis, and built billions of models, from which the four-DNA methylation was identified as prognostic biomarker. And after controlling clinicopathological parameters such as age, tumor stage and tumor tissue location, and so on, it was still proved that this biomarker had a high prediction performance (Figure 3, Figure 3—figure supplement 1 to Figure 3—figure supplement 7). These results indicated that the findings in this study were based on actual signal in methylation data rather than driven by clinical biases.

As for *MITF,* a melanocyte-specific modulator also recognized as a lineage addiction oncogene in melanoma, we are sorry for missing the comparison of our model with *MITF*, which is a favorable prognosticator of OS in melanoma patients (Garraway et al., 2005; Naffouje et al., 2015). According to the reviewer's suggestion, we have compared the sensitivity and specificity of our model and *MITF* expression. The result show that the AUC of *MITF* (AUC: 0.538, 95%CI: 0.42-0.65) was smaller than that of our model (AUC: 0.754, 95%CI: 0.66-0.854). And we also have made changes to Results section and Supplementary file 2.

In the Results section: “In addition, numerous prognostic markers have previously been identified for CM, utilizing archival tumor tissues or single institutional studies. *MITF* has been identified as a lineage survival oncogene amplified in malignant melanoma (Garraway et al., 2005).”

In addition, in our manuscript, the significance level was determined based on *P* values in univariate cox regression analysis with the cut-off of *P* value as 0.001. Here in univariate cox regression analysis, there were only three methylation markers with P less than 1x10^-7^, which was not sufficient for subsequent analysis, so Bonferroni correction was not implemented. But as an alternative, Benjamini and Hochberg False Discovery Rate correction was performed, and the results indicated all of our four methylation sites were also significantly (*P* < 0.001) associated with patient survival. We also made changes to Supplementary file 1 in the revised manuscript.

3) Validation: The four-DNA methylation signature "beats" the other biomarkers in Supplementary file 2/Figure 4A-C. Data may not be available for all of these studies, but for those based on DNA methylation, how does the four-DNA signature compare to the datasets used to generate other methylation predictive markers (Supplementary file 2 17-DNA methylation, CTLA-4, etc.)? Presumably these had validation datasets as well (it seems unfair to expect the four-DNA signature of this study to "beat" the training set for another marker, but other validation sets should be a reasonable comparator). The authors did include a comparison using their validation set, but additional validation datasets would strengthen the generalizability of their biomarker/signature.

This is a valuable suggestion. For other known prognostic biomarkers, on the one hand, if the prognostic model of the biomarker is available in the previous research, we directly calculated the risk scores of patients in our validation dataset using their model, then verified their predictive performance through performing ROC analysis, and the AUCs were obtained, just as the analysis for our four-DNA methylation signature. For instance, the model for the four-lncRNA signature has been reported in the previous studies (Chen et al., 2017). On the other hand, if the biomarker was single gene or DNA methylation site, such as CTLA-4 expression and CTLA-4 methylation (cg08460026), we directly performed ROC analysis to verify their predictive performance using them as test variables in our validation dataset, and the AUCs were obtained. Meanwhile, if the biomarkers are known, but the model is unavailable, such as a 17-DNA methylation signature (Sigalotti et al., 2012), we would first perform multivariate Cox regression analysis to construct model in our training cohort. Then according to the model, we calculated the risk score for each patient in validation cohort, and performed ROC analysis to verify their predictive performance. Ultimately, Supplementary file 2/Figure 4A-C were obtained.

In addition, as suggested by the reviewer, additional validation datasets indeed would strengthen the generalizability of the biomarker. In fact, we have conducted a comprehensive search before the submission of our paper, but only one publicly available dataset (GSE51547) that meets the requirements, which had been used as an additional validation dataset to examine and compare our signature. Recently, we have launched another search in NCBI, National Cancer Database (NCDB), Surveillance Epidemiology and End Results Program (SEER) database, and so on, but other than the validation dataset we used in this study, there is no other publicly available dataset containing both the DNA methylation data and related clinical survival information of patients at present. Undoubtedly, we will make further validation when there are available independent datasets in the future.

4) It would be helpful to include a supplemental figure/panel illustrating distribution of the 4-site methylation risk predictor score value for the TCGA melanoma cohort and then plot the threshold dividing "low-risk" and "high-risk" groups (median). Subsequently, the Figure 3 (main text) and supplementary figure legends need to be explained better (e.g. in Figure 3A "low-risk (28/69)" what does (28/69) mean? I seem to be unable to find this in the text or figure caption.)

According to reviewer’s kind suggestion, we added a supplemental figure (Figure 1—figure supplement 2) illustrating distribution of the four-methylation risk predictor score value in both training cohort and validation cohort, and made changes in the manuscript as follows.

In the Results section: “To determine the potential predictive value of this four-DNA methylation signature in the prognosis, Kaplan–Meier curves along with the Wilcoxon test were used to visualize and compare the OS of patients in the low- versus high-risk group which were classified using the median risk score (3.69) of the training cohort as the cutoff point, and the distribution of the risk predictor scores for the training and validation cohort was illustrated in Figure 1—figure supplement 2.”

We are sorry for not giving the detailed explanation of "low-risk (28/69)" in figure 3 and other figures. In this study, based on the median risk score (3.69) of training cohort, the patients were classified into low- or high-risk groups, low-risk group with lower risk scores, and high-risk group with higher risk scores. In Figure 3A, “low-risk (28/69)” refers to that a total of 69 patients in the low-risk group, in which 28 with last clinical status “death”, and other figures are similar. To make this more clearly in all figures, all figure legends were updated.

5) One additional panel in Figure 1 (or supplement) might be helpful: principal component analysis plot using methylation values at 4 selected biomarkers that illustrates separation between "high" and "low" survival groups. This should clearly illustrate direct effect of methylation level on survival differences.

Thanks for the reviewer’s kind suggestion. Principal component analysis (PCA) was carried out using four methylation values at selected biomarkers. The PCA scores plots exhibited discrimination between short survival (OS < 5 years) and long survival (OS > 5 years) patients. We added a new supplementary figure as Figure1—figure supplement 1.

In the Results section: “Patients exhibiting long-term survival tended to have lower methylation levels of cg24670442, cg18456782 and higher methylation levels of the other two methylation sites, consistent with the results of multivariate Cox regression analysis. Moreover, principal component analysis (PCA) was carried out using four methylation values at selected biomarkers (Figure 1—figure supplement 1). The difference of PC1 and PC4 is 15.42%, indicating the continuous capturing of information. And the combination of four methylation markers can effectively distinguish patients with long- and short-term survival.”

6) Subsection “Association of the four-DNA methylation signature with ICB immunotherapy-related signature”: "significantly negatively correlated with PD-1, PD-L1, PD-L2, CTLA-4 (P < 0.05 […]". Multiple hypothesis testing significance adjustment should be considered. Based on Figure 4D – 4-meth signature correlation was estimated against 7 other signatures (or 6 if GBP5-meth is considered a part of 4-meth), thus significance should be determined using P-value threshold P < 0.05/7 (if Bonferroni correction is used). Please, report raw p-values for this analysis in the supplement.

We apologize for the lack of significance adjustment. According to the reviewer's suggestion, we have adjusted the *P* values by applying the Bonferroni correction and we have added raw *P* values and adjusted P values for this analysis in revised manuscript and Supplementary file 3.

[Editors' note: further revisions were requested prior to acceptance, as described below.]The manuscript has been improved but there are some minor remaining issues that need to be addressed before acceptance, as outlined below:The reviewers appreciate the effort made by the authors to address their concerns. The result is a strengthened manuscript.1) Without another validation dataset (major point #3), the generalizability of the results will remain a potential weakness, but publishing this study with this caveat noted in the text/discussion is of value.It is recommended that the authors mention this, for example by adding the following sentence to the end of the third paragraph of the Discussion section: “When further samples become available it will be important to analyze this methylation signature in another validation dataset.”

Thanks for the reviewer’s kind suggestion. We made changes in the manuscript as follows.

In the Discussion section: “Furthermore, it was demonstrated that in both the validation and independent cohorts, our signature outperformed other known prognostic biomarkers, including mRNA, lncRNA, and DNA methylation, and statistical comparison using Z-test revealed that it has significantly higher (*P* < 0.05) predictive performance than almost all the other known biomarkers. When further samples become available it will be important to analyze this methylation signature in another validation dataset.”

2) The DNA variant in intron of OCA2 has been found to have a protective effect in melanoma GWAS (rs4778138, β = -0.18, PMID: 26237428). The authors state that cg18456782 (OCA2) marker "had positive coefficients, indicating a correlation between higher DNA methylation level and shorter overall survival". If one looks at the TCGA survival data and separates melanoma patients by median OCA2 expression, there is a significant differential survival (p~1e-5) with low expression favoring better survival. This is consistent with the presented data and suggests that while in GWAS the OCA2 variant has been found having a protective function, survival and expression data suggests a risk pattern for OCA2 gene.Discussing this in the Discussion section would strengthen the manuscript because there is no sufficient experimental data on OCA2.

We deeply appreciate the reviewer’s insightful suggestion. We made changes in the manuscript as follows.

In the Discussion section: “*OCA2* is involved in the melanin biosynthetic process and mammalian pigmentation (Crawford et al., 2017), and the DNA variant in intron of *OCA2* (rs4778138) has been found associated with CM risk (Law et al., 2015). The hypomethylation levels of cg18456782 (*OCA2*) was associated with lower expression of *OCA2* and a lower risk. Meanwhile separating CM patients by median expression of *OCA2*, there is a significant differential survival (*P* < 0.0001) with low expression favoring better survival. All these results suggest a risk pattern for *OCA2* gene in CM.”

3) Prior to the analysis the training cohort was not adjusted for clinical parameters like age, tumor stage/grade and tumor tissue site; the authors have explained how they controlled for this, but this fact should be mentioned in the Discussion section so that results could be interpreted with the appropriate caution.

We agree with reviewer’s comments. To make this clear in our manuscript, we made changes as follows.

In the Discussion section: “In order to be clinically useful, a DNA methylation signature must be independent of clinical factors. Here we adopted a grouping method not influenced by our subjective, that is based on the TCGA series number of patients, without adjusting for clinical parameters.”

4) The Abstract should be modified as follows:Cutaneous melanoma (CM) is a life-threatening form of skin cancer. Prognosticbiomarkers can reliably stratify patients at initial melanoma diagnosis according to risk, and may inform clinical decisions. Here, we performed a retrospective, cohort-based study analyzing genome-wide DNA methylation of 461 patients with CM from the TCGA database. Cox regression analyses were conducted to establish a four-DNA methylation signature that was significantly associated with the overall survival (OS) of patients with CM, and that was validated in an independent cohort. Corresponding Kaplan-Meier analysis displayed a distinct separation in OS. The ROC analysis confirmed that the predictive signature performed well. Notably, this signature exhibited much higher predictive accuracy in comparison with known biomarkers. This signature was significantly correlated with immune checkpoint blockade (ICB) immunotherapy-related signatures, and may have potential as a guide for measures of responsiveness to ICB immunotherapy.

Thanks for the reviewer’s improving comments, and we modified the abstract as suggested.